# Marine Bacterium *Vibrio* sp. CB1-14 Produces Guanidine Alkaloid 6-*epi*-Monanchorin, Previously Isolated from Marine Polychaete and Sponges

**DOI:** 10.3390/md17040213

**Published:** 2019-04-04

**Authors:** Tatyana Makarieva, Larisa Shubina, Valeria Kurilenko, Marina Isaeva, Nadezhda Chernysheva, Roman Popov, Evgeniya Bystritskaya, Pavel Dmitrenok, Valentin Stonik

**Affiliations:** G.B. Elyakov Pacific Institute of Bioorganic Chemistry (PIBOC), Russian Academy of Sciences, Prospect 100 let Vladivostoku, 159, 690022 Vladivostok, Russia; shubina@piboc.dvo.ru (L.S.); valerie@piboc.dvo.ru (V.K.); issaeva@piboc.dvo.ru (M.I.); chernysheva.nadezhda@gmail.com (N.C.); prs_90@mail.ru (R.P.); belyjane@gmail.com (E.B.); paveldmt@piboc.dvo.ru (P.D.); stonik@piboc.dvo.ru (V.S.)

**Keywords:** guanidine alkaloids, 6-*epi*-monanchorin, HRESI MS, ^1^H NMR spectra, marine bacteria, *Vibrio* sp., polychaete, *Chaetopterus variopedatus*, 16S rRNA gene analysis, phylogenetic reconstruction

## Abstract

Twenty-three bacterial strains were isolated from the secreted mucus trapping net of the marine polychaete *Chaetopterus variopedatus* (phylum Annelida) and twenty strains were identified using 16S rRNA gene analysis. Strain CB1-14 was recognized as a new species of the genus *Vibrio* using the eight-gene multilocus sequence analysis (MLSA) and genome sequences of nineteen type *Vibrio* strains. This *Vibrio* sp. was cultured, and 6-*epi*-monanchorin (**2**), previously isolated from the polychaete and two sponge species, was found in the cells and culture broth. The presence of the 6-*epi*-monanchorin was confirmed by its isolation followed by ^1^H NMR and HRESIMS analysis. These results showed the microbial origin of the bicyclic guanidine alkaloid **2** in *C. variopedatus*.

## 1. Introduction

Various guanidine-containing natural products, isolated from different marine invertebrates, demonstrate antifungal, antibacterial, antiviral, and antitumor properties [1] and are suitable compounds for drug development due to high levels of their bioactivities and water solubility. Moreover, some natural guanidine-containing compounds, such as streptomycin, have already been introduced in the clinic. 

Most of the marine guanidine alkaloids found in marine sponges are polycyclic, and two bicyclic representatives of this group, namely monanchorin (**1**) and 6-*epi*-monanchorin (**2**), are known. These compounds were isolated from representatives of two phylogenetically distant taxa, namely marine sponges *Monanchora ungiculata* and *Halichondria panicea* (phylum Porifera) [2,3], and marine polychaete *Chaetopterus variopedatus* (phylum Annelida) [4]. Previously it has been reported that the compound **1** shows weak cytotoxic activity against IC2 murine mast cells [2], while compounds **1** and **2** (Figure 1) are able to inhibit the migration and colony formation of cisplatin-resistant cancer NCCIT-R cells [4].

The presence of the alkaloids **1** and **2** in such different taxa could indicate that a common unidentified marine microorganism(s), accumulated in both sponges and the polychaete is a genuine producer of these secondary metabolites of unknown biogenesis. Really, the presence of the same secondary metabolites in distantly related animal taxa sometimes point to potential symbiotic or dietary sources of the corresponding substances. However, experimental evidences of their microbial origin were rarely obtained. Recently, we compared levels of these alkaloid content in different body parts of the polychaete *C. variopedatus* [5,6]. Both alkaloids were predominant into the food net parts of the animals and the content of 6-epi-monanchorin (**2**) was very high (5.0% of dry weight) [4]. These findings prompted us to undertake the present study. We have tried to identify the biogenetic origin of the above mentioned polychaete metabolites. 

## 2. Results and Discussion

### 2.1. Isolation of Microorganisms

The secreted mucous net of the polychaete was pre-rinsed in sterile sea water. Pieces of tissue were aseptically removed and homogenized in sterile sea water. Bacterial strains were isolated by plating samples of tissue homogenates onto medium plates containing the modified MN medium [7]. The plates were incubated aerobically at 20 °C for 7 days. The bacterial colonies that grew on the Difco^TM^Marine Agar 2216 Becton, Dickinson and Company (BD) with that medium were picked up and classified morphologically and biochemically. Twenty-three bacterial strains were isolated in pure cultures and then analyzed by MALDI MS. 

### 2.2. Preliminary Identification of Potential Microorganism-Producers by Monitoring of the Compounds Giving Ion Peaks at m/z 212.17 by MALDI MS, Characteristic of 1 and 2 MS 

A preliminary screening procedure was carried out using Ultrafex III MALDI TOF/TOF mass spectrometer and Biotyper Software (Bruker Daltonics) to select isolates for further analyses. Sample preparation was carried out by “direct transfer” procedure (Ver. 2.0 Biotyper). Spectra were calibrated with external calibration by *Eschercihia coli* DH5 alpha standard and protein calibration standard I (Bruker Daltonics). The majority of identified bacterial strains were represented by *Vibrio* spp. and, thus, *Vibrio* was the dominant group of bacteria cultured from the mucous net of this polychaete. The occurrence of the compounds with a peak at *m*/*z* 212.17 in MALDI MS, presumably corresponding to 6-*epi*-monanchorin or monanchorin (**2** or **1**), are shown in the Table 1. In total there were eleven promising strains found, seven of which were identified as *Vibrio* spp. and gave this ion peak in MS. It should be noted that the data obtained by this method should be considered as preliminary and did not allow accurate identification, neither microorganisms nor target compounds.

### 2.3. Identification of Compounds **1** or **2** by HRESIMS

The authentic identification of compounds **1** and **2** into the three promising strains such as CB1-14, CB2-11, and CB2-6 (see Table 1) was carried out after isolation of these compounds by HPLC followed by analysis with HRESIMS. The strains were incubated at 200 rpm in 100 mL of modified MN liquid medium at 28 °C for 7 days. After incubation, the whole cultures were centrifuged to harvest the bacterial cells. Then cells were suspended in water (30 ml), frozen, and subjected to ultrasonic treatment. The suspension was extracted with EtOAc, and the organic phase was evaporated to dryness. The resulting mixture was dissolved in a small amount of EtOH, and extracts were subjected to HPLC on ODS-A columns. The fractions with retention times of 12 to 17 min were collected and analyzed by HRESIMS. Compounds showing ion peak with *m*/*z* 212.1757 [M + H]^+^ (calcd for C_11_H_22_N_3_O, 212.1757) were isolated from strains CB1-14 and CB2-11. Strain CB1-14 showed a more intense peak at *m*/*z* 212.1757 compared with that for CB2-11. Their mass spectra were identical to the spectra of standards. As a result, monanchorins were identified in these two strains. However, in order to determine which of two epimeric compounds was biosynthesized by these microorganisms, it was necessary to isolate the alkaloids in amounts sufficient for the obtaining of NMR data.

### 2.4. Isolation and Identification of 6-epi-Monanchorin by ^1^H NMR Spectroscopy

The strain CB1-14 was chosen for preparative isolation of target compounds. After incubation of 12 L medium at 28 °C for 7 days, the culture broth of *Vibrio* sp. strain CB1-14 was separated from cells by centrifugation. The 6-*epi*-monanchorin (**2**, Figure 1) was isolated from the EtOAc extracts of both cells and lyophilized culture broth using reverse-phase HPLC. The structure was exactly identified on the basis of ^1^H NMR and HRESIMS data by comparison with authentic sample [4]. As a result, it was found that the CB1-14 strain biosynthesizes 6-*epi*-monanchorin (**2**). Monanchorin itself was not found in this strain in amounts sufficient for NMR spectrum recording. 

### 2.5. 16S rRNA Gene Sequence Analysis of Bacterial Isolates

Twenty bacterial isolates selected for screening for compounds **1** and **2** production were identified by 16S rRNA gene analysis on the EzBiocloud server [8]. Based on the sequence comparison to reference type strains, the isolates were assigned to the two bacterial phyla (Proteobacteria and Firmicutes). Two isolates (CB1-13 and CB1-18) showed the highest similarity values with *Bacillus hwajinpoensis* SW-72^T^ (99.21%–99.24%) and *Bacillus hemicentroti* JSM 076093^T^ (98.27%). One isolate (CB1-3) shared the highest similarity value with *Pseudovibrio japonicus* WSF2^T^ (99.2%), *Pseudovibrio ascidiaceicola* DSM 16392^T^ (98.99%), and *Pseudovibrio denitrificans* DSM 17465^T^ (98.91%) from Alphaproteobacteria. The others were closely related to the species of the genus *Vibrio* from Gammaproteobacteria. The isolates CB1-14, CB2-10, CB2-8, and CB1-5 showed 98.96%–99.58% sequence similarity with *Vibrio hangzhouensis* CN83^T^. The other isolates CB2-5, CB1-7, and CB2-12 had 98.79%–100% sequence similarity with *Vibrio barjaei* 3062^T^ and *Vibrio thalassae* MD16^T^. Most of isolates (CB1-1, CB1-10, CB1-11, CB2-4, CB2-9, CB2-11, and CB2-13) shared the highest similarity values with *Vibrio mediterranei* CIP 103203^T^ (99.65%) and *Vibrio shilonii* AK1^T^ (99.59%). Three isolates (CB1-6, CB2-1 and CB2-7) showed similarity values less than 97.5% with reference type strains of the species of genus *Vibrio*. 

The phylogenetic tree based on the 16S rRNA sequences (1438 bp) clearly showed that *Vibrio* isolates grouped into four clades (Figure 2), three of which included the single type strains, *V. barjaei* or *V. hangzhouensis* or *V. mediterranei*. The fourth clade was at the base of the genus *Vibrio* and did not include any type strains.

Guided by the cutoff value at the species level equal to 98.65% [9] and phylogenetic positions, the isolates CB1-13 and CB1-18 might be identified as *B. hwajinpoensis* and the isolate CB1-3 as *P. japonicus*. Among *Vibrio* isolates, the isolates CB2-5, CB1-7, and CB2-12 might be identified as *V. barjaei*, the isolate CB1-5 as *V. hangzhouensis*, and the isolates CB1-1, CB1-10, CB1-11, CB2-4, CB2-9, CB2-11, and CB2-13 as *V. mediterranei*. The clades, containing CB1-14 and CB2-1, might be distinguished as candidates for new species. Thus, one of bacterial strain (CB1-14) presumably producing monanchorins was identified as *Vibrio* sp., closely related to *Vibrio hangzhouensis* CN83^T^, but probably distinguished from this species. 

Therefore, the phylogenetic analysis revealed bacterial diversity in the mucus net of the *C. variopedatus*. The dominant cultured bacteria were members of the genus *Vibrio*, belonging, at least, to three different species.

### 2.6. Multilocus Sequence Analysis of CB1-14

The phylogenic analysis based on 16S rRNA gene sequences showed the isolate CB1-14 was closely related to *V. hangzhouensis* CN83^T^, sharing 98.96% identity with this strain. It means that the calculated identity value is within the boundary range proposed for delineating *Vibrio* species [10]. Since the 16S rRNA gene sequence did not help in differentiating closely related bacterial species, the eight-gene MLSA was applied as that currently used for delimitating *Vibrio* species [11,12].

To overcome difficulties in application of universal primers for the MLSA, the draft genome of CB1-14 was obtained and used to retrieve sequences of eight housekeeping genes. Following previously described MLSA scheme [11] and using available genome sequences of nineteen type strains including *V. maritimus* CAIM 1455^T^, *V. variabilis* CAIM 1457^T^, *V. mediterranei* NBRC 15635^T^, and *V. hangzhouensis* CGMCC-1-7062^T^, the MLSA study was performed. Based on phylogenies generated by ML (Maximum Likelihood), MP (Maximum Parsimony), and NJ (Neighbor Joining) methods (data are not presented) and split tree decomposition analysis (Figure 3), the MLSA placed the isolate CB1-14 into the Mediterranei clade. Within the clade, the isolate CB1-14 formed a separate branch closely related to *V. maritimus* CAIM 1455^T^ and *V. variabilis* CAIM 1457^T^, with a high bootstrap support.

Thus, the phylogenetic reconstruction showed that the isolate CB1-14 should be recognized as a new species in the genus *Vibrio*. The valid description of this new species in *Vibrio* genus, isolated from an organ of the polychaete, namely from its mucous net, will be done in a special journal.

Marine invertebrates are the oldest animals on Earth, distributed over all the ocean biomes from polar to tropical waters and from shallow to very deep substrates. In the course of their evolution, marine invertebrates have acquired long-term and stable associations with a wide diversity of bacteria, cyanobacteria, archaea, and other groups of microbes, which make up to 60% of the biomass of some these animals and are essential to their survival [13]. There are a number of reports that cultures of microorganisms, isolated from marine sponge [14,15,16,17,18,19] and ascidian tissues [20], produce secondary metabolites previously isolated from these invertebrates, that indicates their microbial origin [13]. However, up to date, only biosurfactants were identified from polychaete-associated microbial isolates [21]. Production of the 6-*epi*-monanchorin by *Vibrio* sp. CB1-14 isolate is completely unprecedented. Most of the compounds so far isolated from *Vibrio* spp. were proved to be non-ribosomal peptides or their hybrids. Only a few guanidine-containing secondary metabolites were isolated from *Vibrio*, for example siderophore vanchrobactin from *Vibrio anguillarum* [22] as well as Na channel blocker tetrodotoxin and its derivatives from bacteria *V. alginolyticus, V. harveyi, V. fischeri*, and *Vibrio* sp. [23,24,25,26].

Our finding that the *Vibrio* sp. CB1-14 isolate, obtained from polychaete food net and which is able to biosynthesize compound **2**, shows that this bacterium (and probably some other close related species) has important unrecognized biosynthetic capabilities, and should be considered as a potential microbial source of monanchorins. From the biotechnological viewpoint, the cultivation of bacteria after optimization of 6-*epi*-monanchorin production could help to solve the recognized supply problem of marine-derived drugs. Identification of this producer also opens prospects of bicyclic guanidine alkaloid biosynthesis. 

## 3. Materials and Methods

### 3.1. General

The ^1^H-NMR spectra were recorded on a Bruker Avance III-700 spectrometer in CDCl_3_. Chemical shifts were referenced to the corresponding residual solvent signal (δH 7.26/δC 77.20 for CDCl_3_). ESI mass spectra (including HRESIMS) were obtained on a Bruker maXis Impact II LC-MS spectrometer by direct infusion in MeOH. MALDI-TOF mass spectra were obtained on a Bruker Ultraflex III TOF/TOF laser desorption spectrometer coupled with delayed extraction using a Smartbeam MALDI 200 laser with α-cyano-hydroxy cinnamic acid as the matrix. HPLC was performed on a Shimadzu instrument with a RID-10A refractive index detector using a YMC-ODS-A (250 × 10 mm) column. 

### 3.2. Animal Material

Three specimens of the polychaete C. *variopedatus* were collected from the coastal waters by scuba at a depth of 6–10 m (salinity 33%, temperature 20) in Troitsa bay, Peter the Great Gulf, Sea of Japan, Russia, in August 2016 and identified by Dr. B. B. Grebnev (GB Elyakov Pacific Institute of Bioorganic Chemistry of Far Eastern Branch of Russian Academy of Sciences, Vladivostok, Russia).

### 3.3. Isolation of Microorganisms 

The secreted mucous net of polychaete were pre-rinsed in sterilized sea water. Pieces of tissue (about 1 g) were aseptically removed and homogenized in 5 mL sterilized sea water. Bacterial strains were isolated by plating samples of tissue homogenate (0.1 mL) onto Petri dishes with the modified MN medium containing 75% natural sea water, 25% distilled water, 0.12 mM CaCl_2_, 0.15 mM MgSO_4_ × 7H_2_O, 0.09 mM K_2_HPO_4_ × 3H_2_O, 8.8 mM NaNO_3_, 0.19 mM Na_2_CO_3_, 0.0013 mM disodium EDTA, 0.014 mM Citric acid × H_2_O, 0.015 mM Ferric ammonium citrate, 1% Bacto agar at pH 8.5. These dishes were incubated aerobically at 20 °C for 10 days. The bacterial colonies that grew on the modified MN medium were picked up and then pure bacterial cultures were grown on Difco^TM^Marine Agar 2216 (BD) and classified morphologically and biochemically. Bacterial strains were stored in 30% glycerol solution at −80 °C.

### 3.4. Identification of Microorganisms by MALDI MS 

Bacterial isolates were stored at −80 °C on the Microbank system (VWR, Darmstadt, Germany). Selected colonies were isolated from plates using a sterile pipette tip and applied directly onto a 384-position ground steel target plate (Bruker Daltonics, Bremen, Germany). The samples were immediately mixed with 2 µL of saturated solution of α-cyano-hydroxy cinnamic acid (HCCA, Bruker Daltonics) in 50% acetonitrile (Sigma–Aldrich, Taufkirchen, Germany), supplemented with 2.5% trifluoroacetic acid (Roth, Karlsruhe, Germany). The matrix/sample spots were crystallized by air drying. Spectra of bacterial strains from the KMM collection of our Institute, as well as reference strains from different commercial collections, were used for comparison. Spectra were calibrated with external calibration by *Escherichia coli* DH5 alpha standard and protein calibration standard I (Bruker Daltonics).

### 3.5. Incubation of Microorganisms for HRESI MS Analysis of Compounds

The bacterial strains (CB1-14, CB2-11, and CB2-6) were incubated at 200 rpm, at 28 °C for 7 days, in the 100 mL liquid-modified MN medium. For preparative isolation of target compounds, the strain CB1-14 was incubated at the same conditions using 12 L of the medium. 

### 3.6. Isolation and Structure Identification of 6-epi-Monanchorin

After incubation, the cells and culture broth of *Vibrio* sp. (strain CB1-14) were separated by centrifugation at 5000 rpm for 30 min. The cells were suspended in water (50 mL), frozen, and, after de-freezing, subjected to ultrasonic treatment. Then the suspension was extracted with EtOAc, and the organic phase was evaporated to dryness. Further chromatographic purification of the obtained residue with reversed-phase HPLC (YMC-ODS-A, 250 × 10 mm) using EtOH-H_2_O (55:45% + 0.05% TFA) gave pure compound **2**.

**6-*epi*-monanchorin** (**2**, 0.2 mg), HRESI MS *m*/*z* 212.1757 [M + H]^+^ (calcd for C_11_H_22_N_3_O, 212.1757). ^1^H NMR (700 MHz, CDCl_3_): 8.74 (1H, br.s, H-2), 8.64 (1H, br.s, H-4), 7.08 (2H, br.s, H-10), 4.84 (1H, t, *J* = 3.0 Hz, H-1), 3.90 (1H, ddd, *J* = 6.0, 1.6, 7.6 Hz, H-6), 3.33 (1H, dt, *J* = 6.0, 1.6 Hz, H-5), 2.36 (1H, m, H-9a), 2.24 (2H, m, H-8), 2.13 (1H, m, H-9b), 1.74 (1H, m, H-11a), 1.52 (1H, m, H-11b), 1.40 (1H, m, H- H-12a), 1.30 (5H, m, H-12b and H_2_-13 and H_2_-14), 0.89 (3H, t, *J* = 6.7 Hz, H_3_-15).

### 3.7. DNA Isolation and Amplification and Phylogenetic Analysis of 16S rDNA Gene

Genomic DNAs from bacterial isolates were prepared using NucleoSpin kit (Macherey-Nagel, Germany) according to the recommendation provided by the manufacturer. PCR amplification of 16S rDNA gene from all the isolates was performed according to [27], using primers BF-20 (5′-ATCACGCGTAAAAATCT-3′) and BR2-22 (5′-CCGCAATATCATTGGTGGT-3′), resulting in about a 1500 bp length PCR product. The purified PCR fragments were sequenced using the ABI PRISM 3130xl Genetic Analyzer (Applied Biosystems) and by the BigDye v.3.1 sequencing kit (Applied Biosystems) (see Appendix A in Appendix A). Obtained sequences were analyzed on the highest percentage of similarities using the Ez-taxon database [8] and the MEGA program version 7 [28]. The 16S rRNA phylogenetic tree was constructed using the maximum likelihood (ML) method based on the Tamura 3-parameter model [29], with 1000 bootstrap replications in the MEGA program.

### 3.8. Genome Sequencing and Multilocus Sequence Analysis of CB1-14

A draft genome sequence of the isolate CB1-14 was obtained using 454 GS Junior (Roche Life Science, USA). A *de novo* assembly was performed using Newbler version 3.0 software Junior (Roche Life Science, USA). The genome sequence was assembled into 621 contigs with 14,322 bp of N50. The estimated genome size was 5.3 Mb. Gene prediction and automated genome annotation were carried out using RAST v. 2.0 with default parameters [30]. Sequences of eight protein-coding genes (*ftsZ*, *gapA*, *gyrB*, *mreB*, *pyrH*, *recA*, *rpoA*, and *topA*) from twenty taxa were retrieved from the CB1-14 draft genome, and from the GenBank/DDBJ/EMBL databases. The MEGA program was used to concatenate, align, and reconstruct the ML, maximum parsimony (MP), and neighbor-joining (NJ) phylogenies with 1000 bootstrap replications. The best-fit model for protein evolution determined in the MEGA program was HKY+G [31]. Split decomposition analysis was performed using SplitsTree version 4.14.3 with a neighbor net drawing and a Jukes–Cantor correction [32,33].

## 4. Conclusions

Our results present the first evidence of the microbial origin of 6-*epi*-monanchorin (**2**), previously isolated from the secreted mucus trapping net of the marine polychaete *C. variopedatus*. Using the 16S rRNA gene analysis, it was revealed that diverse *Vibrio* species are dominant bacteria cultured from the *C. variopedatus* mucus net. In addition, it was shown that these bacteria belong to several different species of the genus *Vibrio*. The strain CB1-14, producing alkaloid **2**, was recognized as a new species in the genus *Vibrio* by phylogenetic reconstruction using eight protein-coding genes. Our results suggest that filter-feeding polychaetes should be considered as a novel source of alkaloid-producing bacteria.

## Figures and Tables

**Figure 1 marinedrugs-17-00213-f001:**
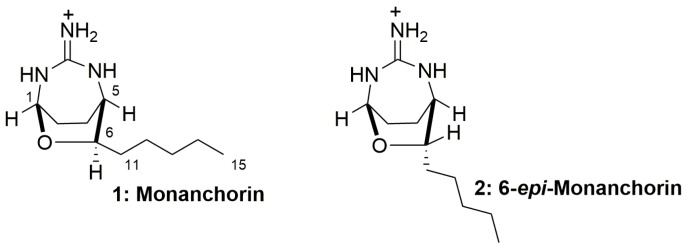
Structures of natural compounds **1** and **2**.

**Figure 2 marinedrugs-17-00213-f002:**
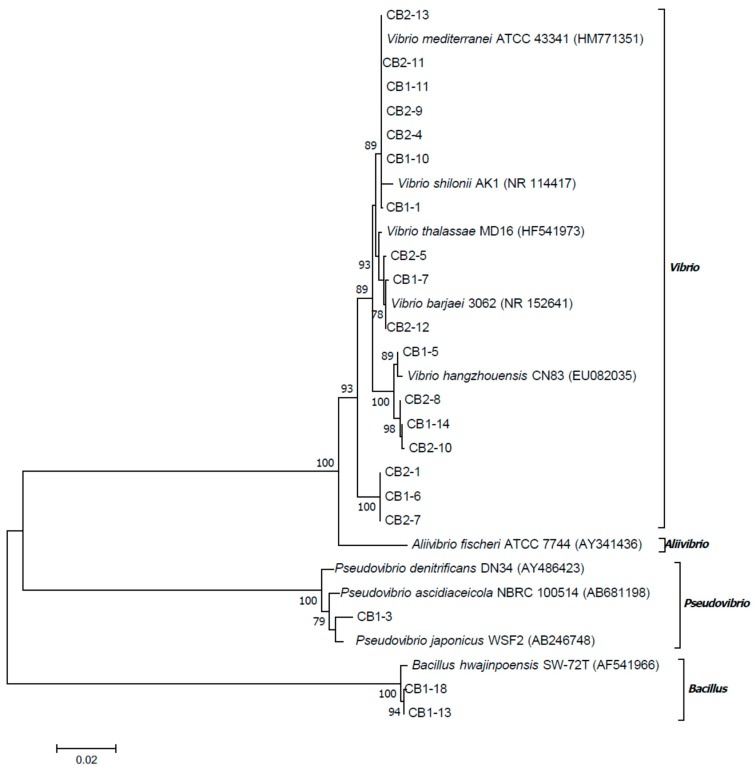
Bacterial phylogenetic tree on the basis of 16S rRNA gene sequences of isolates recovered from the mucus net of the *C. variopedatus* and closely related sequences of type strains. The tree topology was obtained using the maximum likelihood method based on the Tamura three-parameter model. Bootstrap values above 75% calculated from 1000 re-sampling are shown on the node. The scale bar represents the number of substitutions per site.

**Figure 3 marinedrugs-17-00213-f003:**
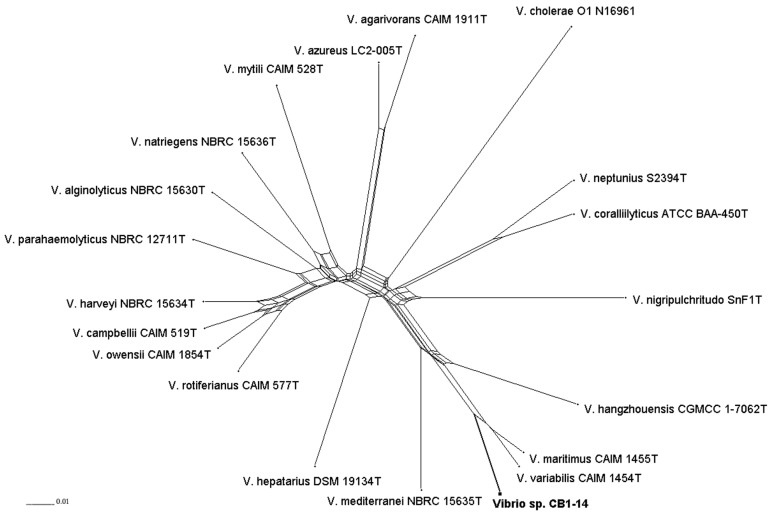
Concatenated split network tree based on eight gene loci. The *ftsZ*, *gapA*, *gyrB*, *mreB*, *pyrH*, *recA*, *rpoA*, and *topA* gene sequences from 20 taxa were concatenated including the isolate CB1-14 (bold font). Phylogenetic tree was generated using the SplitsTree4 program.

**Table 1 marinedrugs-17-00213-t001:** Taxonomic position of microorganism-producers and occurrence of compounds with *m*/*z* 212.17 ion peak by MALDI MS data.

No	Strain	Taxon	*m*/*z* 212.17 *	No	Strain	Taxon	*m*/*z* 212.17
1	CB1-3	nd	nd	13	CB2-3	nd	nd
2	CB1-5	nd	nd	14	CB2-4	*Vibrio* sp.	nd
3	CB1-6	nd	nd	15	CB2-5	Vibrio sp.	nd
4	CB-1-7	*Vibrio* sp.	nd	16	CB2-6	Vibrio sp.	+
5	CB1-8	nd	nd	17	CB2-7	nd	ad
6	CB1-9	nd	ad	18	CB2-8	*Vibrio* sp.	ad
7	CB1-10	*Vibrio* sp.	ad	19	CB2-9	*Vibrio* sp.	ad
8	CB1-11	*Vibrio* sp.	ad	20	CB2-10	nd	ad
9	CB1-12	nd	nd	21	CB2-11	*Vibrio* sp.	+
10	CB1-13	nd	nd	22	CB2-12	*Vibrio* sp.	ad
11	CB1-14	nd	+	23	CB2-13	nd	nd
12	CB2-1	*Vibrio* sp.	nd				

nd, not detected; ad, ambiguous detected; +, detected.

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
