# Peer review of "Marine Bacterium *Vibrio* sp. CB1-14 Produces Guanidine Alkaloid 6-*epi*-Monanchorin, Previously Isolated from Marine Polychaete and Sponges"

_marinedrugs, 2019, doi:10.3390/md17040213_

Round 1

Reviewer 1 Report

General impression

The authors present the first evidence of the microbial origin of 6-epi-monanchorin (2), previously isolated from the secreted mucus trapping net of the marine polychaete C. variopedatus. Using the 16S rRNA gene analysis, it was revealed that diverse Vibrio species are dominant bacteria in the C. variopedatus mucus net. Among these Vibrio species, the strain CB1-14, producing compound 2, was recognized as a new species in the genus Vibrio by a phylogenetic reconstruction. However, the authors briefly described the production of compound 2 from CB1-14 cells. A more quantitative evidence of the microbial origin of 6-epi-monanchorin is required.

Problems to be fixed

The authors seek to characterize the microbial origin of 6-epi-monanchorin from the Vibrio sp. (strain CB1-14). After incubation in the liquid modified MN medium, the cells and culture broth of Vibrio sp. (strain CB1-14) were separated by centrifugation. The cells were disrupted and extracted with EtOAc. Further chromatographic purification of the obtained residue with reversed-phase HPLC. Usually, microbial metabolites are accumulate in the medium as the cells proliferate. To confirm the production of 6-epi-monanchorin from the strain CB1-14, quantitative data (or HPLC histogram) from the CB1-14 cells and culture supernatant are required.

Minor issues

Spell check required: page 2, line 47 “H. panacea”; page 3, line 83 “inMS”; page 4, line 108 “polycahete”.

For the 16S sequence, the sequence should be submitted to the NCBI or provided in supporting information.

Author Response

a) We have included HPLC chromatograms of extracts from the CB1-14 cells and culture supernatant to quantitatively confirm the production of 6-epi-monanchorin by this microorganism, as shown in Supplementary Materials.

b) We accepted all the Reviewer #1 minor corrections.

c) We have provided all 16S rRNA gene sequences to the NCBI and the obtained GenBank accession numbers included in Table S1 of Supplementary Materials.

We are very grateful to Reviewer #1 for his comments about our paper.

Reviewer 2 Report

The review entitled “Marine bacterium Vibrio sp. CB1-14 Produces Guanidine Alkaloid 6-epi-Monanchorin, Previously Isolated from Marine Polychaete and Sponges” by Makarieva et al. identify a Vibrio sp. as the producer of a natural product originally isolated from its’s host, a polychaete.

The main methodological approach is based on microbial methods and sequencing, for example the definition of Vibrio sp. as a new ‘species’. This is not my main area of expertise and I would recommend to the editor to invite a microbiologist to judge this part of the work. I find it curious that the authors claim to find at least three distinct species of bacteria (1-14 and 2-11) and possibly 9 more isolated from the same bacterial cultivation which are supposed to produce the same natural product. This would be a major finding – 3 individual species of bacteria associated with the same host produce the same secondary metabolite. I find it much more likely that several genotypes from the same species were isolated from the tissue. Therefore, I wonder if the phylogenetic differences between the 11 analyte-producing bacteria are not because of methodological issues of the 454 sequencing. However, if carefully checked and proven this conclusion would have to be brought into the foreground. I would recommend to provide the raw sequencing data to the microbial reviewer with the revision of this ms. The authors also claim, “that diverse Vibrio species 281 are dominant bacteria in the C. variopedatus mucus net”, which is not supported by the data.

Secondly, I would ask the authors to provide a supplementary section of the raw spectra and analytical data (1H NMR, HRMS) so that the chemical analytical work can be reviewed – I’m sure the authors confidently identified the compound of interest but this can’t be judged at present based on the data provided, especially because the presented data (1 H in CDCl3 at 700 MHz) is not consistent with the method section (500 MHz MeOD).

Lastly, it is my personal opinion that the authors sell their results short. Their main conclusion in the end – polychaetes are interesting sources for bacteria with natural products – pretty much is true for any xenic system. A few additional experiments and a wider discussion could find a much larger audience and achieve increased impact. I find the presented study does not add any new information from a ‘drugs’ perspective – although the editor should decide if the outcomes warrant a publication in this journal.

From a chemical ecological perspective, however, the authors have shown an intriguing model system of a polychaete-bacteria-holobiont that is worth exploring (although I’m well aware of my bias – being a marine chemical ecologist). The fact that the authors could isolate and identify the natural product from the host’s tissue in the earlier study, suggests that the producing bacteria and/or its metabolite must occur in significant abundances (an interesting finding even if all bacteria turn out to be the same species). This could be visualized with a targeted FISH probe, for example, to illuminate the spatial bacterial association within the host’s organ/tissue. At the minimum a sequencing of the microbiome community should be attempted to show relative abundance differences of bacteria from different genera in the host tissue. The authors also have standards available that should enable a targeted quantification per tissue weight and area –giving insights in the analytes’ concentration. The intriguing following discussion would highlight the resilience (?) of the host towards the cytotoxic effect. Does the analyte harm the host? Is it a pathogen? Is it a mutualistic bacterium? How does the host fare when the bacteria are removed by antibiotic treatments? It is remarkable that it is found in sponges also, again – symbiont or detrimental bacterium?

While I don’t suggest that these elements are required for the publication I would like the author’s to consider broadening their readership.

A few detailed comments below:

Line 77. 6-epi-monanchorin or monanchorin (1 or 2): 6-epi-monanchorin is defined as 1, the same as in the earlier publication. The later text refers to 6-epi-monanchorin as 2 (e.g. 246, 284).

Line 82: Revise table legend.

Table 1: I do not understand the ‘- -‘ and ‘+ -‘ classification. Printer error?

Line 86: Justification is missing. Why are CB1-14, CB2-11, and CB2-6 considered promising from 11 strains identified by MALDI? Chosen at random?

Line 102: “The strain CB1-14 was chosen for preparative isolation of target compounds”. Why?

Line 111ff: Why is one of the ‘promising’ strains, CB2-6 missing from the phylogenetic analysis?

Line 147-149: This section needs a lot more critical discussion of the results. Do the authors claim that these bacteria are separate species that are all associated with the same host AND all produce the same secondary metabolite? That would be a HIGH impact finding if true - but I find it easier to believe that this is rather due to methodological sequencing characteristics.

Line 199: Not consistent with results section.

Line 202: Please provide the mass accuracy for this instrument

Line 210: Dr. Grebev is not a co-author of this ms and missing from the acknowledgments, oversight?

Line 241: something’s not right here, you sonicated a frozen samples? Freeze dried?

Line 245: Impossible. One HPLC step on C18 from raw bacterial pellet gives a ‘pure’ compound? Please provide spectra.

Line 281: Claim not supported by the data. Vibrio was dominant of the cultivated bacteria with large inherent culture bias. No sequencing community data of the host shown in this ms.

Author Response

We would like to indicate that one of coauthor (Valeria Kurilenko) is a high level microbiologist.

1a. We consider that our major finding is that the Vibrio sp. CB-14 produces 6-epi-monanchorin. This conclusion was done by us only after large-scale cultivation of Vibrio sp. CB-14 and isolation of the compound 2 by HPLC in amounts sufficient for the obtaining of NMR data. However, we did not find, that 3 individual species of bacteria associated with the same host produce the same secondary metabolite. Compounds showing ion peak with m/z 212.1757 [M+H]+ from the strain CB2-11 might be either monanchorin (1) or other isomeric compounds with the same molecular formula C11H22N3O. In addition, we have shown that the strain CB2-6 does not contain any compounds with molecular formula C11H22N3O (see section 2.3. Identification of compounds 1 or 2 by HRESIMS).

1b. We have performed identification of bacteria via direct sequencing of 1,500 bp-PCR fragments of 16s rRNA genes using genomic DNAs isolated from pure bacterial cultures. We did not conduct a metagenome 454 sequencing. The all 16S rRNA nucleotide sequences were determined by Sanger sequencing that is devoid of methodological issues of the 454 sequencing. However, the obtained sequences were screened for chimeras using DECIPHER server (http://www2.decipher.codes/FindChimeras.html). The DECIPHER did not detect any chimeras in the sequences. The data are summarized in Table S1 (Supplementary Materials). Thus, we have not isolated several bacterium genotypes from the same species.

1c. We have provided the raw sequencing data of the bacteria in Table S1 (Supplementary Materials).

1d. We consider that “diverse Vibrio species are dominant”, it is supported from data of Fig. 2 because there are 17 Vibrio isolates among 20 isolates. We have corrected the corresponding sentence in the text.

2. We feel that our presented study adds novel and very important information for the search of drug candidates because we have shown that the bioactive bicyclic guanidine alkaloid 6-epi-monanchorin has the microbial origin. Experimental evidences of microbial origin of marine natural products were rarely obtained. From the biotechnological point of view, the cultivation of bacteria after optimization of 6-epi-monanchorin production could help to solve the recognized supply problem of this marine derived compound because the polychaete C. variopedatus is very rare animal and not good source for isolation bioactive compound in amounts sufficient for drug development.

3. We agree with the referee that there are perspectives for development of this study in the field of chemical ecology. We hope that marine ecologists will take our research into account and with their further research they will be able to answer very interesting questions from the reviewer.

Concerning detailed comments:

Line 77. We have changed “6-epi-monanchorin or monanchorin (1 or 2)” for “6-epi-monanchorin or monanchorin (2 or 1)”;

Line 82. We have revised Table 1 legend.

Table 1. We have corrected Table 1.

Line 86. We have clarified why CB1-14, CB2-11, and CB2-6 strains are promising strains (Table 1).

Line 102. We have clarified why the strain CB1-14 was chosen for preparative isolation of target compounds by addition of the corresponding sentence in the text.

Line 111. The strain СB2-6 was rejected by us after HRESIMS data.

Line 147-149. We consider that the data of Fig. 2 in the text of the manuscript confirm the diversity of Vibrio.

Line 202. The mass accuracy for Bruker maXis Impact II LC-MS spectrometer is about 2.0 ppm. Line 241. We have corrected the sentences containing “frozen cells”.

Line 245. We have experience in isolation of pure compounds when one HPLC step is enough for simple mixtures. We provided the corresponding spectra in Supplementary Materials.

We are very grateful to Reviewer #2 for his comments about our paper.

Reviewer 3 Report

This is a very interesting manuscript submitted for publication by Makarieva et al., reporting the bacterial production of a guanidine alkaloid previously isolated from a marine sponge and a marine worm. The procedures have been very adequately performed and the results are sounding. Bacterial identification and phylogenetic analysis is also relevant.

Considering the importance of the findings, the authors should seriously perform the following improvements of the manuscript and reported investigations:

a) unfortunately the English language is extremely poor. The authors should send the manuscript to one professional service of scientific language checking, such as the American Journal Experts, in order to provide a much better written text for publication.

b) the authors should include pictures showing the HRMS and 1H-NMR spectra of both the standard and the bacterial guanidine, in order to better illustrate their findings.

c) the authors should attempt to perform incorporation experiments with either 1-[13C]acetate, or 2-[13C]acetate, or 1,2-[13C]acetate and report these results in the same manuscript. These experiments will very significantly add relevance to their findings.

Author Response

a) We have sent the manuscript to native English speaking colleague to provide a much better written text for publication.

b). We have included pictures showing the HRESIMS and 1H-NMR spectra of both the standard and the bacterial guanidines, in order to better illustrate our findings.

c). We will think about a performing of incorporation experiments with either 1-[13C]acetate, or 2-[13C]acetate, or 1,2-[13C]acetate in future. In addition, we hope to obtain full genome sequence of discovered by us marine Vibrio sp. (CB1-14) to establish genome sequences connected with biosynthesis of guanidine alkaloids.

We are very grateful to Reviewer #3 for his comments about our paper.

Round 2

Reviewer 1 Report

The authors modified the paper according to most suggestions, turning it suitable for publication. 

Author Response

We are very grateful to Reviewer #1 for comments to  the manuscript.

Reviewer 2 Report

RE: 2nd review of the paper “Marine bacterium Vibrio sp. CB1-14 Produces Guanidine Alkaloid 6-epi-Monanchorin, Previously Isolated from Marine Polychaete and Sponges” by Makarieva et al..

Some of the findings presented in this paper still need to be improved before publication. As recommended in my first review, the authors need to make the sequencing results available for the reviewer. The GenBank accession were made available in the supplementary section but the record has not yet been released. The authors should either make the data available or release the GenBank. EU082035, 3062T and CIP103203T used for the 16 comparison, show more than 97% similarity, thus the author’s claim that the dominant cultured bacteria were members of the genus Vibrio, belonging, at least, to three different species needs to be shown.

Line 17: The authors want to publish the species identification in a separate journal. This is fine, but the definition of CB1-14 as a new species should phrased more carefully here.

Line 73: dominant group of bacteria cultured from

Line 282: are dominant bacteria cultured from the C. variopedatus mucus net

286: Isn’t it ‘before unexplored’?

Author Response

We have inserted <The GenBank accession numbers for all 16S rRNA nucleotide sequences will be available on online NCBI server after July 20 2019 (gb_submitted_answer.doc)> to the text in SM part (Line 379). We would like to present personally the data to the reviewer #2 as an additional file (NCBI_bacterial isolates_16S.txt). To show that all 17 Vibrio strains are the members of the genus Vibrio, we corrected the figure 2 in the text by including a type strain of the Genus Aliivibrio most closely related to the Genus Vibrio. The bootstrap value of 100% clearly showed belonging them to the Genus Vibrio. Also, we included other closely related Vibrio species to figure 2. An analysis of the 16S rDNA tree showed that most strains were belonged to three known species of Vibrio.

Line 17: We have corrected the phrase about the strain CB1-14 recognized as a new species of the genus Vibrio.

Lines 73, 282 and 286: We have accepted all the Reviewer #2 minor, but very important corrections.

We are very grateful to Reviewer #2 for some new comments to improve the manuscript further.

Sincerely yours

Dr. Tatyana Makarieva

Round 3

Reviewer 2 Report

congrats